# Consumer preferences, experiences, and attitudes towards telehealth: Qualitative evidence from Australia

**Kaylie Toll**[1]*, **Lauren Spark**[1], **Belinda Neo**[1‡], **Richard Norman**[1], **Sarah Elliott**[1,2‡], **Leanne Wells**[2‡], **Julia Nesbitt**[2‡], **Isobel Frean**[3‡], **Suzanne Robinson**[1,4]

**1** School of Population Health, Faculty of Health Sciences, Curtin University, Perth, Western Australia, Australia, **2** Consumers Health Forum of Australia, Canberra, Australian Capital Territory, Australia, **3** Digital Health CRC Limited (DHCRC), Sydney, New South Wales, Australia, **4** Deakin Health Economics, Deakin University, Burwood, Victoria, Australia

☯ These authors contributed equally to this work.
‡ BN, SE, LW, JN and IF also contributed equally to this work.
* kaylie.toll@curtin.edu.au

**Data Availability Statement:** We have provided demographic data within the manuscript. The KTD interview schedule are now provided as a supplementary information file to allow the study to

## Abstract

In Australia, telehealth is not new, with several telehealth specialist services being available for those living in rural and remote communities. However, prior to the COVID-19 pandemic, telehealth was not routinely available for primary care or urban specialist appointments. There has been an increased focus in the use of telehealth within primary care, and particularly general practice, but overall, there has been limited research to date to guide telehealth best-practice based on consumer experiences and preferences within these settings. We aimed to capture the consumer experience of telehealth during the COVID-19 pandemic, through a novel Kitchen Table Discussion (KTD) method. This increases access to a broader community consumer cohort, with consumer hosts leading discussions in a safe environment. The KTDs were conducted in May 2021, with 10 community members each hosting a group of up to 10 participants. A total of 90 participants took part from across Australia, with the majority living in major cities, although a significant proportion lived in inner and outer regional areas of Australia, or had experience living in rural, regional or remote areas. Seventy percent of participants reported using telehealth in the past. Data were analysed sequentially using thematic analysis and identified key themes: modality, convenience, access, wait time, existing relationship, communication, connectivity, cost, and privacy. Overall, the future of telehealth looks hopeful from the perspective of the consumer, but significant improvements are required to improve consumer engagement and experience. It is evident that 'one size does not fit all', with results suggesting consumers value the availability of telehealth and having choice and flexibility to use telehealth when appropriate, but do not want to see telehealth replacing face-to-face delivery. Participants tended to agree that telehealth was not a preferred method when physical examination was required but would suit certain points of the patient journey.

be duplicated. Due to ethics and data governance policies, we are not able to provide the interview transcripts. Data are available from the Curtin University Ethics Committee (contact via hrec@curtin.edu.au) for researchers who meet the criteria for access to confidential data.

**Funding:** This research was part of a larger study supported by the Australian Government Department of Health, Health Economics and Research Division, and the Digital Health CRC Limited (DHCRC), funding number DHCRC0161. The DHCRC is funded under the Commonwealth's Cooperative Research Centres (CRC) Program. The Department of Health had no role in study design, data collection and analysis, or preparation of the manuscript.

**Competing interests:** The authors have declared that no competing interests exist.

# Introduction

Telehealth is not new in Australia, with well-established services operating across rural and remote communities for many years. The COVID-19 pandemic has hastened the spread of digital solutions across the country and now casts telehealth into a new light [1], with its appeal of allowing health services, including general practice and allied health, to be accessed by people in their own homes.

Whilst there are multiple definitions, telehealth is defined here as a method of delivering healthcare services via information and communication technologies (ICT), transmitting audio, images and data between a patient (consumer) and healthcare provider [2]. These services can be used in diagnosis, treatment, preventative and curative healthcare.

Internationally, telehealth services grew exponentially in response to the COVID-19 pandemic, rapidly moving online and adapting new models of care to maintain continuity of care [3]. Despite the reported positive outcomes and experiences [4–7], many services have reverted back to similar pre-COVID levels, with spikes in usage as areas go in and out of lockdown.

In Australia, a number of telehealth specialist services were available for those living in rural and remote communities, including psychiatry from 2002, specialist and consultant physician services (2011), allied mental health services (2017), and some GP services (2019), however, pre-COVID telehealth was not routinely available for primary care or urban specialist appointments. The Medicare Benefits Schedule (MBS) is a listing of items where the Australian Government will provide financial assistance towards the cost of public medical services [8]. Temporary MBS telehealth items were introduced in repsonse to COVID-19, to "allow people to access essential Medicare funded health services in their homes and reduce their risk of exposure to COVID-19 within the community" [9]. As at June 2021 in Australia, telehealth services now respresent 18% of all MBS items, increasing from the pre-COVID level in March 2020 of 7.9%, but well off it's peak of 35.6% in April 2020 [10]. The modality of this telehealth usage is overwhelming via the telehone, 93% compared to 7% via videoconference [10]. A. The Australian Government has enabled provision of both phone and video consultations, with video being the preferred substitution for face-to-face when telehealth is clinically appropriate. However, uptake for phone consultation is higher and it is unknown if these usage rates are due to consumer preference or the availability given to consumers.

Given the longevity of telehealth implementation, there is a substantial body of telehealth research pre-dating the COVID-19 pandemic. However, much of this work focuses on rural and remote and/or hospital settings. Since 2020, there has been an increased focus in the use of telehealth within primary care, and particularly general practice, but overall, there has been limited research to date to guide telehealth best-practice based on consumer experiences and preferences within these settings. Such information is key to informing the development of policy, sustainable funding incentives, and rigorous frameworks to ensure the quality and safety of health care delivered digitally [11].

A number of telehealth reviews have drawn on research pre-COVID, there are also publications that draw on small case studies, and a vast range of commentary and perspective pieces [3, 12–15]. Most of these publications relate to telehealth as a video consultation mode, with very limited work around telephone consultations. In addition, the majority of research and commentary is from the clinical and policy perspective, with limited focus on consumer perspectives within primary care and general practice settings, both in Australia and globally.

Although descriptive and extensive across settings, the research to date particularly within the primary care setting lacks a strong evidence base, whereby benefits and concerns of consumers and clinicians have largely been perceived rather than based on strong evidence or experience. The lack of the lay input from consumers in the current literature narrows the

applicability of these findings and its translation for use in general telehealth practice [11]. Studies that have focused on consumer experience and preference have tended to do so in a unidimensional way that does not capture the complexity of consumer preferences [16]. There is a need to rigorously explore consumer experiences, preferences, and attitudes towards telehealth in the primary care setting, to ensure consumer engagement and sustainability of the health system [17].

This paper aims to fill this gap in knowledge by capturing the consumer experience of telehealth during the COVID pandemic. It does this through the use of a novel method, the Kitchen Table Discussion, that can access a broader community consumer cohort and includes consumers leading discussions in a safe environment that welcomes in-depth conversation.

## Methods

Kitchen table conversations have been used in various settings and contexts such as local government and political campaigns [18–21]. The Kitchen Table Discussion (KTD) methodology, pioneered first in the consumer sector by Health Consumers Queensland [22], supports the training of consumer facilitators (hosts) with a strong community network to conduct focus groups with peers. The sessions involved discussing consumer attitudes and experiences via a semi-structured focus group format. The aim of this approach is to enable informal and relaxed dialogue, so health consumers, carers and community members, who do not ordinarily participate in healthcare consultations, are able to have their say in a safe and supportive environment.

### Recruiting and training the hosts

The KTD hosts were selected via purposive sampling through a widely disseminated expression of interest process coordinated by The Consumers Health Forum of Australia (CHF). Applications were assessed on the basis of a potential host's community links and the level of diversity this could deliver in terms of KTD participants.

Ten trained consumer hosts (facilitators) were selected for the role, required to identify participants through their community links and were aware that their participant group should represent diversity as far as possible. Hosts received: training via a Zoom videoconference; a comprehensive host guide; questions to ask participants during the discussion; a feedback report template; support from CHF to ensure a successful session; and appropriate remuneration in recognition of their time. This training and guidance were designed to emphasise their role as a moderator, and not a participant or responder.

### Participants and recruitment

Participants were recruited by their hosts through non-probability convenience sampling with each host inviting up to 10 community members via their own local community connections, to attend either online or in-person group discussions, and to achieve a sample size allowing for a variety of experience and options to emerge. The participants were community acquaintances of the host, enabling health consumers, carers and community members who do not ordinarily participate in healthcare consultation to have their say in a safe and supportive environment. The recommendations to hosts for sampling from the host guide was as follows: "The group of people should reflect a wide diversity of ages, cultures, and health experiences. Please choose your participants mindfully, ensuring there is no conflict of interest, or you feel coerced in any way." Due to illness one KTD host held individual consultations with participants by phone or online. Participants were required to be 18 years or older.

We adopted a two-stage consent process, participants were able to consent to the study and consent to the audio recording separately. Not consenting to the recording did not exclude participants from the study.

Each participant received an information sheet outlining what the project entailed, who was doing the research, how the research would be used, and who would have access to their de-identified information. Participation was voluntary, and participants could withdraw from the study at any stage. The participants were notified they would be asked to verbally answer some questions within their discussion group and provide some written basic demographic information regarding gender, age, chronic conditions, and if they had experienced telehealth previously.

Participants were advised that the questions in the survey were designed not to cause any distress. If the participant felt anxious about any of the questions, they did not need to answer them. Strategies were put in place to support and assist any participants who may feel sensitive or upset after hearing some experiences of healthcare, and to please let the host know if this occurs.

Participants were given a $60 gift voucher to reflect the value of their time and for participating in the study.

## Conducting the focus groups

Hosts were given a step-by-step guide to run their sessions, and a feedback report template containing a set of 10 questions, plus additional question prompts (available in **S1 File**).

In short, participants were asked how and why they may have accessed telehealth services in the past year and discussed their experiences. Based on consumer experiences, common themes explored:

- The types of telehealth used across services;

- Circumstances where telehealth was used;

- The benefits of telehealth in reducing geographical isolation;

- The convenience of telehealth;

- The importance of an existing consumer-clinician relationship for satisfaction and confidence;

- The importance of a positive first experience with telehealth to inform future engagement; and

- Reasons why telehealth was not used.

Each participant was given a chance to answer each question, using their own words, including real-time check back by hosts to ensure the information captured was true to the respondent. Participants were invited to make any additional comments at the end of each session or email them following the session. Hosts recorded participant responses to each question onto the template. The sessions were conducted between 10[th] and 28[th] May 2021 and lasted approximately 90 minutes each and were either recorded and transcribed, or scribed during the session. The host guide recommended to audio record each session to allow better engagement in discussion and support report template completion. Some participants did not consent to the audio recording, which did not exclude them from the study, and other groups encountered technical difficulties.

The hosts completed a written report of participant responses based on a template and attached demographic data forms, participant registration sheets and completed consent

forms, and sent to the research team. A member of the research team transcribed all available electronic recordings. All but one KTD session was conducted in English. Whilst we did not target non-English-speaking focus groups, in selecting KTD hosts we particularly set out to ensure diversity in terms of the potential hosts' community links. One host had strong links to the Japanese community thus contributing to inclusion of participants from culturally and linguistically diverse communities. Because of the variability in English language proficiency of participants the KTD was undertaken in Japanese. The host translated the outcomes into English.

## Data analysis

The qualitative analysis employed a combination of inductive (generating new knowledge) and deductive (testing theories) techniques [23]. All transcribed session data, report templates and demographic data forms were transferred to researchers electronically, making sure all data was de-identified. Data were analysed using thematic analysis and identified key themes.

The phases of thematic analysis outlined by Braun and Clarke [24] were applied by the research team, in the six following steps:

- Familiarisation with the data

- Generation of initial codes

- Searching for themes

- Reviewing themes

- Defining and naming themes

- Producing the report.

Data were analysed sequentially, for each of the 10 questions that comprised the semi-structured focus group interview template followed by the hosts.

For each question, an independent researcher analysed the transcriptions and host reports through an open-coding method to generate initial codes. These codes were then applied, reviewed, and categorised into sub-themes, each then defined and labelled with a collective theme that described the core concept for each question. To ensure rigour and validity a second researcher independently reviewed the transcripts, the coding and categorisation of subthemes.

Quotes are used to demonstrate findings in each theme from the voice of the participant. Each quote includes the participant allocated identification number, 1–90, location based on the Modified Monash Model and past telehealth usage. For example: P5, major city, used telehealth.

Ethics approval was obtained from Curtin University Human Research Ethics Committee (HREC number HRE2021-0232).

## Results

A total of 90 participants took part in the KTDs, with between 7 to 10 participants attending each KTD (mean = 9). Table 1 shows the majority of the participants were female (79%; n = 71). Most participants were aged between 35 to 54 (42%; n = 38), followed by those aged 55 to 74 (32%; n = 28), aged 18 to 34 years (18%; n = 16), with 9% (n = 8) aged over 75.

Approximately half of participants were living with a chronic health condition (47%; n = 42). This included one group largely made up of participants with kidney disease, from those on dialysis to end-of-life care. Other groups included people from diverse backgrounds,

**Table 1.  Summary of participant demographic characteristics.**

| Variable | Participants (n = 90) | Proportion (%) |
|---|---|---|
| **Gender** | | |
| Female | 71 | 79% |
| Male | 17 | 19% |
| Other | 2 | 2% |
| **Age** | | |
| 18–25 years | 2 | 2% |
| 25–34 years | 14 | 16% |
| 35–44 years | 18 | 20% |
| 45–54 years | 20 | 22% |
| 55–64 years | 14 | 16% |
| 65–74 years | 14 | 16% |
| 75–84 years | 7 | 8% |
| >84 years | 1 | 1% |
| **Living with a chronic health condition** | | |
| Yes | 42 | 47% |
| No | 48 | 53% |
| **Experience using telehealth** | | |
| Yes | 69 | 77% |
| No | 21 | 23% |

including low socio-economic areas, Aboriginal and Torres Strait Islander communities, and non-English speaking. One group was made up of native Japanese speakers and was held in Japanese, the host translated responses into English. More than two thirds of participants reported using telehealth in the past (77%; n = 69), with nearly three quarters (69%) conducted over the phone versus videoconference (31%).

A total of 10 KTDs were held across the country, in all states except for the Northern Territory and Tasmania. Most KTDs were held exclusively face-to-face (60%; n = 6), with two held exclusively via Zoom (20%), one held via a combination of face-to-face and Zoom (10%), and one held via phone and Zoom (10%). Face-to-face sessions were held in metropolitan Melbourne, Brisbane, Adelaide, Sydney and Canberra. Two face-to-face sessions were held exclusively in outer regional areas (including: Western Australia and New South Wales). Sessions held via Zoom included participants from Perth and inner regional Western Australia (South West); Brisbane and inner and outer regional Queensland (North); Canberra, Adelaide and Sydney; Melbourne and inner regional Victoria.

Overall, the majority of participants were currently living in major cities, although a significant proportion lived in inner and outer regional areas of Australia, or had experience living in rural, regional or remote areas.

## Modality

Telehealth consults were provided across the public, private and community services sector, with the majority conducted over the phone. The majority of phone consults were with GPs, whereas specialists and hospital-based appointments tended to use videoconference. The majority of reported consultations were with GPs. For GP consults, the primary reasons were for acute or chronic illness, questions around medications or prescription renewals, a follow-up after tests or scans, or to request for a referral to a specialist. Some telehealth consults were linked with a hospital or specialist out-patient appointment. A small number were linked to

allied health (speech, occupational therapy, physiotherapy), nursing or to mental health support (psychologists, psychiatrists).

Participants largely agreed that telehealth was ideal for simple, routine or non-acute situations, where a "diagnosis" is not needed. However, some participants reported being able to send photos to their clinician to support diagnosis and treatment. Participants reported that telehealth is not suited for complex or sensitive issues or where a physical examination is required. Participants reported that the lack of ability to do a physical assessment is often perceived as the inability to treat the consumer comprehensively, holistically or with a preventative focus, such that important things could be missed that may otherwise be detected face-to-face.

"*A biggest failure of telehealth is lack of physical assessment. Can't check your temperature, ear canal, inside the mouth, can't use the stethoscope to check my lungs, can't feel skin rashes, check tooth decays, or any other physical ailments.*"

*P39, major city, used telehealth*

"*I did not feel like some of the services that I had were able to offer the treatment. Like psychology just does not work without eye contact for me, and particularly not when being asked to do some difficult work.*"

*P9, major city, used telehealth*

Telehealth was reported as fit-for-purpose for those living with chronic conditions where telehealth provided an efficient means to maintain ongoing continuity of care for everyday issues.

"*Over the past year I have accessed a phone consultation with my GP and that was very simple. I just wanted a prescription renewed. That was very easy, and she had a bit of chat as to how I was, to cover a standard consultation time.*"

*P1, major city, used telehealth*

When asked about their preference of choice between phone or video there was a relatively equal split between what would have been chosen, with a small number still preferring face-to-face. There were some comments that video would be preferred if needed to inspect anything visually, for hospital appointments, and generally the preference when seeing allied health practitioners. Participants noted that phone may be preferred for GPs as it is quicker and easier than video, with more flexibility and less technical requirements.

A number of participants reported not being given the option between face-to-face and telehealth. One consumer mentioned they needed to request telehealth when initially only face-to-face was offered. Some participants reported perceptions that whether they were offered telehealth options or not depended on the preference of the clinician, and the decision was not consumer-led.

## Themes

Participants were asked about their views on the key benefits, barriers and enablers to engaging with telehealth. Key themes include: convenience, access, wait time, existing relationship, communication, connectivity, cost, and privacy. These themes are outlined in Fig 1 and explored in detail below.

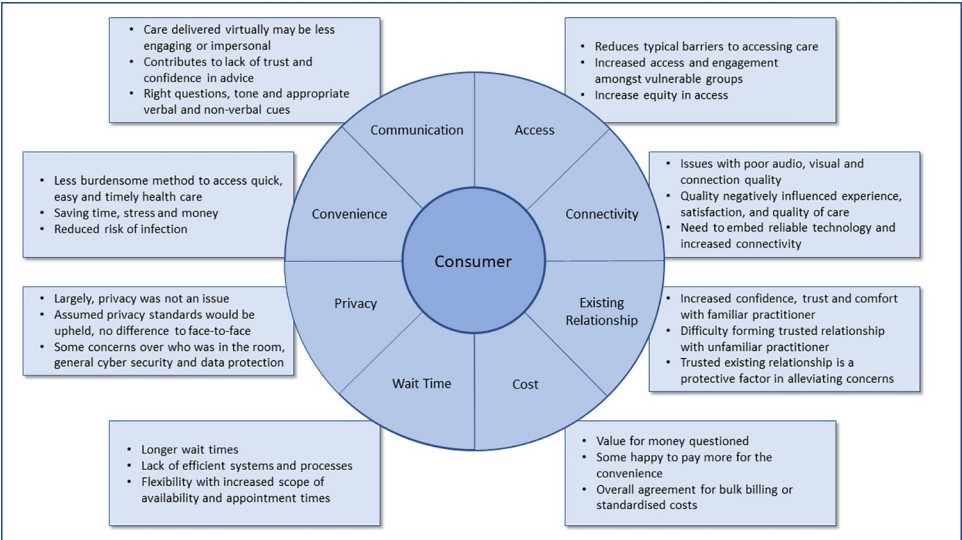

**Fig 1. Illustration of the eight core themes relating to the benefits, barriers, enablers, and future improvements of telehealth.**

**Convenience.** The main strength of telehealth was associated with the overall convenience and efficiency of telehealth, providing a less burdensome way to access quick, easy and timely health care whilst saving time, stress and money. This convenience was also associated with less risk of sharing or being exposed to contagious infections compared to face-to-face care.

Participants reported that telehealth appointments were overall less burdensome, being able to remain at home with less rushing, less stress, and less need to plan and schedule attending in-person.

*"I think it really can't be overstated how difficult travelling to certain appointments can be, even if they are regular standing ones. I think if you have a good relationship with your telehealth provider, then there are some services I am not keen to go to back to in person because it works just fine from my living room."*

*P9, major city, used telehealth*

*"My mobility is limited by pain. My condition makes me frightened getting there. . . makes it difficult, parking difficulties, access, obstacles, logistics, energy resources. Telehealth is fantastic as I do not have to worry about all that."*

*P15, major city, used telehealth*

Some participants reported telehealth overall being quicker and easier than face-to-face, both from their perspective and perceived perspective of practitioners.

*"Definitely being able to get quick check-ins without having to wait at the medical centre, particularly for things like scripts or test results."*

*P46, major city, used telehealth*

Reduced practical logistics associated with attending a face-to-face appointment was a commonly reported advantage, particularly for those with additional needs or responsibilities.

"*Taking large number of kids (five kids) to a clinic is challenging for poor mothers. Especially for Aboriginal communities. There isn't anything to entertain the kids at clinics. Kids get bored and make noise, cry and run off at times. Crèches are not available unless it's a kid's hospital.*"

*P42, major city, used telehealth*

Saving time was the most commonly reported advantage of telehealth. It was common for participants to report overall efficiency in receiving medical care via telehealth, particularly related to time saved not having to travel or bother about traffic and parking or having to sit in waiting rooms for extended periods of time. Participants reported the advantage of being able to engage in other activities while waiting or if the practitioner was running late.

"*The convenience of being able to make an appointment even if I'm not able to make it physically in the office, so there are more options for when I can see the doctor if I don't have to go there physically. Also not having to waste time sitting in a doctor's office when the doctor is running late, I just kept going with my work while waiting for them to call–so much better!*"

*P49, major city, used telehealth*

Fig 2 reports a range of situations in which telehealth was preferred as it provides more convenience and can mitigate a number of risks.

**Access.**   Increased access to care was seen as a common advantage of telehealth, particularly in relation to accessing rapid medical attention with an efficient exchange of information. This was experienced across both acute care (e.g., emergency telehealth) and primary care (e.g., GPs).

Participants reported telehealth has enabled them to access medical care or services that they would otherwise not have available or would otherwise be less likely to engage in should they need to attend face-to-face. This is particularly true for those living in rural and remote

- Avoid risk of acquiring an infection or infecting others if contagious (including COVID-19);
- Reduced time and cost associated with travel, within both metro and more remote areas;
- No need to arrange childcare;
- Easier to attend appointments for the elderly, disabled or vison impaired;
- Easier to attend when sick or injured;
- Easier for a carer or support person to be involved in consults;
- A desirable modality for mental health consults;
- Easier access to a toilet if needed;
- Encourages more independence for those living with additional needs;
- Increased access to after-hours care;
- Less waiting time;
- To reduce waiting lists;
- Can help with ease and timeliness of appointments;
- Reduce risks for clinicians associated with home visits; and
- More privacy and less confrontation to discuss issues over telehealth.

**Fig 2. Participant views on convenience and risk reduction associated with telehealth consultations.**

communities, particularly when they may only have typical access to visiting locums or specialists based in metropolitan areas or interstate.

> "*Telehealth is just so convenient and has improved my access to services that I may otherwise have neglected or have been difficult for me to access.*"
>
> P51, *major city, used telehealth*

Increased access and reduced costs were reported to be a benefit for those living in regional or remote areas. Living in more isolated areas also increased reliance on videoconferencing capabilities where they may be limited phone reception. Participants also reported the flexibility of arranging and attending a telehealth appointment, with an increased scope of availability/appointment times to access medical care in a timely manner.

Participants reported that telehealth should be used to support multidisciplinary approaches to healthcare, providing a more integrated and consumer-focused service. Access should be offered via both video and phone, with consumers empowered to choose their preference. Participants reported that telehealth should be available 24/7 and should be extended beyond GPs and hospitals and further into primary care (e.g., mental health and dietetic services).

> "*Preference comes into it, but I do not want to go back to a world without accessing telehealth. I think for most people it has been good to have that option.*"
>
> P9, *major city, used telehealth*

Participants suggested that telehealth needs to be more accessible to a wider range of people. Participants suggested that engagement and research should explore how telehealth can be maximised among targeted and vulnerable groups, including youth, elderly, homeless, those with a disability, those with a mental health illness, carers, those with a low socio-economic status, and Indigenous communities.

Participants suggested that interpreter services need to be embedded within telehealth, easily and widely accessible to support the Culturally and Linguistically Diverse (CALD) community. Some participants who spoke English as a second language reported discomfort when conversing over the phone and felt telehealth exacerbated communication issues. To compensate for language limitations, these participants reported often relying on visual information (such as viewing of test results), but the lack of this kind of visual informational aid when using the telephone contributed to a reduction in the level of understanding. Video consultations may help to improve these situations, but some participants reported face-to-face contact was still preferred for optimal communication. One suggestion is to explore how telehealth could be used to link consumers to practitioners that speak their language.

It is also important to focus attention on how telehealth could be maximised within urban areas, not just rural and remote.

> "*I think it is great. I'd like to see more telehealth because, as you said, it is better for regional people, people with disabilities, maybe some people who are not comfortable face-to-face. Maybe this is more comfortable for them. The government needs to look more into this, so we don't have these issues with communicating...*"
>
> P5, *major city, never used telehealth*

Some participants raised concerns over digital literacy and what support patients would have with technology issues.

"*These days people are expected to use a digital device to do everything. However, it is unclear which government division is responsible to support those with limited IT literacy and don't have a family member to assist them.*"

*P64, major city, never used telehealth*

**Wait-time.** Participants reported the flexibility of arranging and attending a telehealth appointment, with an increased scope of availability and appointment times to access medical care in a timely manner. The flexibility and increased access of a telehealth appointment compared to face-to-face was associated with the ability to seek medical attention more rapidly.

"*I was able to wait for the appointment in the comfort of my home, rather than a waiting room and didn't have to travel when feeling really unwell or try and find someone to take me.*"

*P53, major city, used telehealth*

However, some participants suggested that telehealth was associated with longer wait times, and an overall lack of efficient systems and processes. This was particularly a deterrent when practitioners were running late and there was poor communication to consumers to advise of the delay. Some participants reported being given a window of a few hours in which their telehealth appointment would be conducted, with others agreeing they often waited for long periods for their appointment to start, unsure if they had been forgotten.

"*And on the few occasions I've used it, one occasion they were extremely late, one occasion they were 15 minutes late and a couple of times they were just about bang on time. So you just have to juggle your time and wait for it unfortunately and that is a weakness.*"

*P19, inner regional, used telehealth*

The lack of perceived respect for consumers' time reflected by practitioners' timely attendance to telehealth consults was a concern for participants.

Overall, while it may be easier and quicker to access care via telehealth, the perceived quality of care of telehealth may be impacted by long waiting times and the inefficiency of current communication systems and processes.

**Existing relationship.** When discussing appropriateness and quality, participants noted the importance of having existing relationships with a trusted and familiar clinician. Participants reported increased confidence, trust and comfort when using telehealth with a regular clinician who was familiar with them and their medical history. There were concerns or hesitations in having a telehealth consult with a new practitioner, reporting either experienced or perceived difficulty forming a trusted relationship and subsequently questioning whether the medical advice provided via telehealth could be trusted.

"*At the time I wasn't visiting my local GP practice and so I was concerned that I would just get any other doctor and they wouldn't know my medical history or anything like that. I know they would provide, you know, the sort of same level of care but I think it's just familiarity with your doctor and already having that established doctor-patient relationship.*"

*P56, outer regional, used telehealth*

Whilst there were concerns that something may be 'missed' via telehealth, even with a familiar practitioner, the presence of a trusted existing relationship may be a protective factor in alleviating such concerns.

"*I do not necessarily trust the online diagnosis; I would like to have met the doctor in person in advance.*"

*P13, major city, used telehealth*

"*. . .I chose to see the doctor in person. I usually see the doctor in person to develop rapport and relationship. I wouldn't trust it over telehealth. I won't be able to open up.*"

*P74, inner regional, never used telehealth*

**Communication.** Overall, poor communication was a commonly reported telehealth weakness, particularly how care delivered virtually may be less engaging or impersonal, and contribute to a lack of confidence and trust in the advice provided via telehealth. Participants identified the importance in the interpersonal and communication skills of the clinician to ask the right questions, in the right tone, with appropriate verbal and non-verbal cues such that consumers feel comfortable and that clinicians are attentive.

"*Medical providers also need to be trained so that the best experience can be enjoyed, with good outcomes. Not just video conferencing protocols and etiquette, but acquiring suitable equipment, for example, cameras that can angle to see injuries or wounds, zoom in and out, large enough screens so providers can see patients' body language, physical condition.*"

*P86, outer regional, used telehealth*

Participants also reported the importance of telehealth consults not being rushed, there needs to be enough time and good communication between consumer and practitioner to exchange information freely, with adequate opportunity for consumers to ask questions. Practitioners also need to have the ability and time to access historical medical records for consumers to feel confident, comfortable and adequately cared for.

"*I think that doctors tend to prioritise in-person appointments and fit the telehealth ones in around that—they don't seem to respect the validity of phone appointments as much and often seem to call you much later than your appointment which then isn't always a convenient time. I had one doctor call me a couple of hours after my appointment and I was in Coles [supermarket] doing the shopping with my kids and he just started talking to me about my test results without checking if it was a good time to talk.*"

*P50, major city, used telehealth*

Good communication, access to adequate medical records, and the ability to ask questions were important factors contributing to whether telehealth provides value for money. Participants noted that a weakness of telehealth is the loss of interpersonal connection and how this could impact on the quality of care.

"*I think that having that interpersonal connection when you are using body language and you are modulating your tone of voice to the person physically in the space with you, makes a big difference. So that is a general weakness that I feel.*"

*P9, major city, used telehealth*

**Connectivity.**    Connectivity and accessibility to telehealth were key barriers, such that poor technical quality greatly influenced the experience, satisfaction and quality of care provided via telehealth. Issues with poor audio, visual and connection quality were commonly reported, particularly for videoconferencing.

"*I was on a phone call telehealth the other day and because the reception in this area of (place-name) is awful, just really awful, my phone call dropped five times over an hour and I had no way of being able to contact the provider back because it was a private line. So each time it would drop out, I had to wait for them to call back. It was just atrocious.*"

*P9, major city, used telehealth*

"*You lose your conversation, you get frustrated. One of you will hang up and then you've got to go back through the rigmarole of getting another appointment. OK if it's only something minor like you've broken your arm and you go to the hospital, fine. But if you've got something major, like you're having a heart attack or you've popped your eye out or something really serious, telehealth's not going to help you.*"

*P18, inner regional, never used telehealth*

Participants were particularly concerned that telehealth may be rolled out on a larger scale as standardised care before adequate investment in quality and stability of services; quality should be setting the pace, not the technology.

Enhancing the quality of virtual interactions through embedding reliable technology and increasing connectivity may improve the consumer experience and value of telehealth, particularly in rural and remote areas.

"*The hospital it was "clunky" never worked most of the time. So, 50% of the time we used telephones (they were meant to be via video). Video consult offers best value and next to face to face consults, so I was bummed when the video didn't work. They would call me, and we have problem with video and going to call via telephone.*"

*P73, major city, used telehealth*

**Cost.**    Participants agreed telehealth was beneficial in terms of reduced costs related to travel, parking, accommodation, and time compared to a face-to-face, especially for those living in regional and remote areas. Participants were asked if they would be willing to pay the same amount or more for a telehealth consultation as a face-to-face consultation.

The majority of participants agreed that telehealth should either be the same or less than a face-to-face consultation. The value for money of telehealth is perceived as lesser than face-to-face, largely because of shorter consultation times, the inability to be assessed comprehensively, and the impersonal nature of a virtual consult contributing to a sense of lower satisfaction in the quality of care provided.

"*Most telehealth appointments are very short, such as less than 5–10 minutes. A telehealth appointment never takes as long as 15–20 minutes. Hence, paying the same price as for a face-*

*to face-consultation is unfair. Lack of physical assessment means the paying the same amount is unfair."*

*P37, major city, used telehealth*

*"I was charged the same fee as the face-to-face consult, which I am not about, because the doctor did not do any physical assessment and I didn't feel that I received the same quality of care as a face-to-face consult."*

*P76, major city, used telehealth*

Some participants may be happy to pay more for the convenience that telehealth affords, although overall there was agreement that telehealth should be fully covered by the public sector MBS or costs at least standardised.

*"I would pay the same but it depends on the situation. If it was urgent and got me medical care that was otherwise unavailable, then I may be happy to pay more. If it was for a minor issue I would expect to pay less."*

*P32, inner regional, never used telehealth*

*"It was very convenient to speak to doctor on phone to organise referral to appropriate specialist. Also had time in consult to talk about vaccination and other issue. Another telehealth service provided a convenient way to get a diagnosis of a simple condition that had re-occurred. Each telehealth appointment was bulk-billed which made it cheaper than a normal appointment."*

*P27, major city, used telehealth*

**Privacy.**    During the discussions we also asked participants two questions relating to privacy and disclosure:

a.  Concerns about the privacy of telehealth; and,

b.  If clinicians have ever discussed privacy issues with them.

A large proportion of participants reported privacy not being an issue. These participants had a standard expectation that health professionals would operate within ethical and professional boundaries.

*"Not really any concerns as I trust it is as private as a face-to-face consult."*

*P45, major city, used telehealth*

Where participants did have concerns, this was primarily related to being unsure who was in the room or could overhear the conversation (from both the consumer and health professional side), and general cyber security and data protection concerns, including hackers.

The majority of participants indicated that privacy had never been explicitly discussed between clinician and consumer. Of those that did recall a discussion, this was normally done at the beginning of a session, if another person was in the room, or general privacy information was provided via a letter or email. Communication generally identified that the same privacy and confidentiality standards that applied to face-to-face applied to telehealth.

A number of participants reported they assumed privacy standards would be upheld, that there were no differences when compared to face-to-face, and that they expected the clinician to advise if otherwise or if any concerns.

"*The doctor was really good and before we started asked if I had a private, comfortable place to chat before we started and she reassured me that she was in her office and it was private and asked me if I had any questions before the consult began–it may be because she knows I live in a share household, but she was great and I wish all my doctors and health professionals took that kind of care.*"

P48, *major city*, *used telehealth*

## Discussion

Overall, the future of telehealth looks hopeful from the perspective of the consumer, but significant improvements are required to improve consumer engagement and experience. Whilst telehealth is a simple modem of communication (technology) it operates in a complex service delivery model, both on the demand and supply sides. Going forward it is important to understand this complexity (which is dynamic and emergent), when implementing telehealth technology into health systems [25]. Results from our study suggest that consumers do not want to see telehealth completely replacing face-to-face consultations, but do value the availability of telehealth and having choice and flexibility to use telehealth when it's appropriate for them. Other studies had reported that the convenience of telehealth mostly outweighed concerns about lack of physical examination when the interaction was generally routine and there was an existing relationship with the provider [5, 6]. Thus, telehealth does have a place in the health system in both primary and secondary care settings, but this needs to be considered in relation to patient-centred health and the care journey as the patient transitions across the continuum of care.

Deciding when telehealth is appropriate across the patient journey is difficult given the complex and multi-dimensional nature of consumer preferences and behaviours [16]. Our research does uncover some interesting insights that go some way towards helping to understand telehealth's place across the health system. Firstly, it is evident that **'one size does not fit all'**. For example, we have participants who reported that not having to present for a face-to-face consultation was advantageous whilst others suggest that they prefer face-to-face consultations. Participants tended to agree that telehealth was not a preferred method when physical examination was required. The lack of ability to do a physical assessment is often perceived as the inability to treat the consumer comprehensively, holistically or with a preventative focus, such that important things could be missed that may otherwise be detected face-to-face. Other studies note that physical examination was seen as an important challenge [5, 6, 26], with some consumers' perception being that without the opportunity to undertake a physical examination, the quality of care would be impacted.

When discussing appropriateness and quality, participants noted the importance of having existing relationships with a trusted and familiar clinician. Like other studies our respondents identified that established patient-practitioner relationship through prior face-to-face contact is important for effective telehealth consults [26]. Like all types of consultations, participants judged success in terms of how well they felt listened to, the manner and behaviour of the clinician, and the clinician's knowledge around their medical history and condition. Other previously published studies also suggest that those who had an established relationship prior to using telehealth felt more comfortable during the conversation [27]. Gordon et al [6] and

Holmström et al [28] both reported that the inability of patients to see their health care provider in person affected the opportunity to engage and build a relationship, also more prominent in telephone consultations where the provider can be heard but not seen [29].

The notion of existing relationships relates to the continuity aspects of good primary care which is important across time, settings, conditions and people. The recent Australian Primary Health Reform Steering Group [30] suggest that primary health should be the lynchpin for continuity of care across all stages of life. Our findings suggest that telehealth incorporated into an effective model of primary care could support this. For example, participants noted that telehealth was appropriate for some routine consultations such as repeat prescriptions, referral or follow up appointments and for some minor acute issues.

Telehealth's convenience and its ability to support accessibility for patients to visit their preferred GP, can support continuity and better self-care management of chronic conditions, with shorter, more frequent telehealth visits, for example supporting in medicine management or reassuring and coaching patients around self-care. Noting there may be need for allowances to the 'usual medical practitioner' rule, where consumers prefer to maintain privacy from a regular or family GP.

The increased scope of availability and appointment times allows for the accessing of medical care in a timely manner. Addressing issues early and within primary care could increase the focus on 'right care, right time, right place'. Study participants suggested that telehealth was not appropriate for more complex or sensitive issues or where a physical examination is required. Similarly, there was mixed opinion regarding mental health consults, with some preferring face-to-face for more severe cases and eye contact, but others preferring telehealth due to the ease of access.

The above highlights the appropriateness of telehealth at certain points of the patient journey. For example, for patients with a chronic condition there are instances along the patient journey that are more suited to telehealth than others. These would include telehealth use in monitoring and 'checking in' with patients, providing medication review, repeat prescriptions or test results, but less appropriate when the patient needs to be examined or have discussions around diagnosis and management of the condition, particularly during the onset of disease. Other studies also noted the convenience of telehealth as a major advantage with respondents noting how having consults through telehealth allowed them to fit consults around 'their day' rather than the providers' day [5]. Donaghy et al [31] reported that the convenience of telehealth is particularly noted for those "who commute to work and for those who have lives structured around work, study, or childcare", and/or live remotely with individuals not needing to book leave from work or pay additional travel costs. Both studies reported that the time reduction had a positive impact in reducing stress and anxiety [5, 31].

It is also worth noting the differences in consumer preferences in regional Australia compared to those in urban and metropolitan areas, with access to primary care and general practice often being more difficult in the former. Telehealth reduces typical barriers to accessing care, particularly related to the practical logistics of attending a face-to-face appointment. Telehealth provides means of increased access and engagement amongst geographically isolated consumers and its implementation could well be different in rural and remote communities. Whilst face-to-face is often the preferred service, if that is not easily available then telehealth provides a real alternative. If implemented well it could support increased access to primary care for these in regional Australia which are home to some of our most vulnerable communities.

An area of importance highlighted in the literature is that of disadvantaged and minority groups. This study had some representation from CALD and Aboriginal and Torres Strait Islander people. There was some indication that for these groups, accessing care via telehealth

can remove some of the stigma and discomfort consumers may experience attending face-to-face appointments, particularly among vulnerable groups. Some consumers reported that not having the stress of travel and sitting in the waiting room was a positive experience for them.

## Technology and connectivity

There are a number of technological aspects that can be challenging for telehealth delivery and uptake in Australia. It goes without saying, that if a patient cannot access the internet or telephone services, then they are not practically able to utilise telehealth services. There is limited consistency in the platforms used and participants suggested the current platforms are "clunky", with common experiences in servers, platforms and websites being down. Communication systems are sub-optimal, particularly if sessions are running behind, with participants often left waiting for a long time for their appointment to start, unsure if they have been forgotten.

Living in more isolated areas also increased reliance on videoconferencing capabilities where they may be limited phone reception. For some individuals telephone has been preferred due to the lack of access and/or knowledge and competence in using video platforms. There is some evidence to say that this is more pronounced for some groups, including older people and people with disabilities [32–34]. This highlights the importance of digital literacy and supporting consumers in increasing their digital literacy skill so there is not inequity in relation to the access and use of telehealth technology.

## Limitations

Whilst this study and its findings provide areas worthy of consideration, the study limitations should be noted. The majority of participants were female (79%), thus there is an underrepresentation of males in the study sample. Bias may be present in the method of convenience sampling and may not be representative of the population. Given the importance of ensuring access and equity to vulnerable groups the lack of representation of CALD and Aboriginal and Torres Strait Islander people within the KTD needs to be addressed in future research. In addition, the study only focused on only one side of the coin and further research that explores clinician's perspectives especially those in primary care (including Allied Health) is important in shaping our understanding of the implementation of telehealth in primary care and the broader health system. A further limitation was the reliance on Hosts to take notes during some sessions when audio recording was not an option.

## Conclusion

The findings reflect the consumer experience of telehealth at a unique point in time where the COVID-19 pandemic had accelerated its public financing and utilisation. Nonetheless, with the pandemic serving as a catalyst to considering how telehealth should be embedded and sustained in healthcare, the findings are instructive. As the pandemic persists, policy makers have several considerations if the community is to see telehealth services continue. Continuation needs to occur in a manner that avoids low value for money, and that provide high value technologies and experiences of care.

The KTD methodology resulted in several key themes, but also made it clear there was not a 'one size fits all' solution. What also remains unclear is the complexity of these consumer preferences in relation to telehealth or face-to-face consults across the spectrum of virtual care services. For example, if virtual care services required a smaller out of pocket cost, would they become a more attractive option? Or would virtual care become an attractive option if it prevents significant travel, and can be conducted with a doctor that the patient knows?

## Policy, practice and research implications

Policy and practice implications must be considered in terms of both the direction of travel in health care, that is, consumer expectations of personalised medicine, integrated care and care close to home, as well as the available policy 'levers' through which change can be achieved. Work needs to be undertaken in the implementation of telehealth and the normalisation of telehealth mode of delivery into the patient journey. Telehealth services should not be full substitutes for face-to-face care, and ideally linked to provision by a provider with a prior relationship with the patient. Exceptional circumstances where it is not possible for patients to have an existing relationship with a particular provider should be catered for to ensure equity of access. For example, 24 temporary MBS telehealth items were introduced in July 2021 for sexual, reproductive health and blood borne viruses which are exempt from the 'usual medical practitioner' rule, allowing greater confidentiality for the consumer [8]. Enhancing the quality of virtual interactions, embedding reliable platforms and technology, and meeting the access and equity needs of vulnerable and targeted groups will go a long way in boosting the confidence consumers have to engage and trust telehealth services.

Telehealth has the potential to support with routine management for those with chronic conditions, but the complexity of the consumer and health services needs to be considered when developing service models. Whilst telehealth provides an opportunity to empower patients and promote consumer choice, the quality of the consult (telehealth or otherwise) can influence consumer satisfaction and perceived effectiveness.

Participants noted the lack of video conference options being offered by general practice and that availability tended to be driven by health professionals rather than consumers. Other studies also highlighted that the availability and modalities of telehealth are often provider driven rather than consumer led [35]. Consumer and patient involvement in the design, delivery and evaluation of telehealth services is crucial to success. Ultimately, consumers should be empowered to choose why, when, where and how they use telehealth services and as such the approach of 'nothing about me without me' should be incorporated into future telehealth policy, practice and research.

## Supporting information

**S1 File. KTD questions for consumers.**
(PDF)

## Acknowledgments

The research team would like to thank the participants, Kitchen Table Discussion (KTD) Hosts and Survey Engine for their time and enthusiastic participation in the research. Without the support and passion of our Hosts and consumer participants this research would not have been possible. We would like to thank staff at CHF and Curtin for their assistance in undertaking this work particularly in organisation of the KTDs and commitment to the project, and the Curtin Research Ethics committee for their detailed and timely response. Finally, we would like to thank the Editor and Reviewers of our manuscript for providing essential feedback allowing for a more precise and well-thought through version of this manuscript.

## Author Contributions

**Conceptualization:** Richard Norman, Leanne Wells, Isobel Frean, Suzanne Robinson.

**Data curation:** Lauren Spark.

**Formal analysis:** Lauren Spark.

**Funding acquisition:** Richard Norman, Suzanne Robinson.

**Investigation:** Julia Nesbitt.

**Methodology:** Richard Norman, Suzanne Robinson.

**Project administration:** Kaylie Toll, Sarah Elliott, Julia Nesbitt, Suzanne Robinson.

**Resources:** Leanne Wells, Julia Nesbitt.

**Supervision:** Richard Norman, Suzanne Robinson.

**Validation:** Suzanne Robinson.

**Visualization:** Kaylie Toll, Lauren Spark, Sarah Elliott.

**Writing – original draft:** Kaylie Toll, Lauren Spark, Belinda Neo, Richard Norman, Suzanne Robinson.

**Writing – review & editing:** Kaylie Toll, Lauren Spark, Leanne Wells, Julia Nesbitt, Isobel Frean, Suzanne Robinson.

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
