## [Decision Letter · Decision Letter 0]

16 May 2022

PONE-D-22-09068Consumer preferences, experiences, and attitudes towards telehealth: qualitative evidence from Australia.PLOS ONE

Dear Dr. Toll,

Thank you for submitting your manuscript to PLOS ONE. After careful consideration, we feel that it has merit but does not fully meet PLOS ONE’s publication criteria as it currently stands. Therefore, we invite you to submit a revised version of the manuscript that addresses the points raised during the review process.

We look forward to receiving your revised manuscript.

Kind regards,

Simone Borsci, Ph.D.

Academic Editor

PLOS ONE

Journal Requirements:

 “This research was supported by the Australian Government Department of Health and the Digital Health CRC Limited (DHCRC). DHCRC is funded under the Commonwealth’s Cooperative Research Centres (CRC) Program.”

“This research was supported by the Australian Government Department of Health and the Digital Health CRC Limited (DHCRC). DHCRC is funded under the Commonwealth’s Cooperative Research Centres (CRC) Program.”

 “This research was supported by the Australian Government Department of Health and the Digital Health CRC Limited (DHCRC). DHCRC is funded under the Commonwealth’s Cooperative Research Centres (CRC) Program.”

Additional Editor Comments :

Dear Authors, the comments of the Reviewers are overall positive. Nevertheless, one of the reviewers provided a list of critical points that need to be adjusted to ensure the replicability of the study.

Reviewers' comments:

Reviewer's Responses to Questions

**Comments to the Author**

1. Is the manuscript technically sound, and do the data support the conclusions?

Reviewer #1: Yes

Reviewer #2: Yes

2. Has the statistical analysis been performed appropriately and rigorously? 

Reviewer #1: N/A

Reviewer #2: N/A

3. Have the authors made all data underlying the findings in their manuscript fully available?

Reviewer #1: Yes

Reviewer #2: Yes

4. Is the manuscript presented in an intelligible fashion and written in standard English?

Reviewer #1: Yes

Reviewer #2: Yes

5. Review Comments to the Author

Reviewer #1: 1. The inclusion of minority groups in this study is key a strength. Could you elaborate if conducting a focus group in a language other than English was intended?

2. What were the experiences of the CALD group in accessing telehealth and were there barriers due to language? Could you elaborate on whether telehealth was conducted with clinicians that spoke the same language as participants or with interpreters or caregivers?

3. It is noted that face-to-face consultations were preferred by participants when physical examinations were required. Was there any evidence of physical examinations being conducted via videoconference? Did participants find this of value and did they feel prepared and comfortable to undertake this in their homes?

4. Is it worth commenting on the under representation of men as a study limitation?

5. Minor typographical errors for review. Line 66: 'response' spelt incorrectly. Lines 72-76: sentence is difficult to read and would be helped if simplified? Line 167: missing 'to' after responses? Lines 211-213: colon after Sydney should be a semicolon or comma? Line 227: missing closed bracket after physiotherapy?

Reviewer #2: Thank you for asking me to review this manuscript. I would firstly like to congratulate the authors on an interesting and timely study. Given the proliferation of telehealth, capturing the consumer experience of telehealth during the COVID pandemic, is indeed a useful exercise.

I do have some suggestions for further improvement of the manuscript:

• Research methodology - Within qualitative, there are number of methodologies (grounded theory, phenomenology, ethnography etc). What is the qualitative methodology for this study?

• Kitchen Table Discussion – I have not heard of this before but sounds very interesting. Why was this chosen as the method of data collection? Has this been tried before in other similar research?

• Sampling – How were the participants identified and recruited? There is no reference in this section. Sampling framework needs more details and references.

• Recruitment – There are two groups here from what I understand. There are the “hosts” and “participants”. It would be good to break the recruitment strategies into these two groups and report it as such. For example, how did the hosts source the contact details of the participants? How did the researchers ensure there was enough coverage of participants? If the host and participants were social acquaintances, how was coercion avoided?

• Data collection – So the hosts recorded participant responses each question onto template and then it was audiorecorded and transcribed? Is that right? If so, why did you collect the same data in two different ways?

• Who did the transcription of the interviews?

• How was observation data managed?

• Rigour –It would be useful to outline clearly the strategies used and how this contributed to the rigour of the data collection and analysis. This is not clearly articulated.

• Data collection –Was there any follow up to clarify any issues? How did member checking actually occur? Again, there needs to be references here to defend your choice.

• Sample size – how was sample size determined?

• Sometimes you use the word “interview” and sometimes it is focus group. Please be consistent.

• How did you ensure rigour and trustworthiness?

• “The interviews opened by asking respondents about the modality of telehealth services and which if any services are preferred for telehealth” – this is Method right? Not results.

• “Telehealth consults were provided across the public, private and community services sector, with the majority conducted over the phone.” – where is the data to back this up in text?

• For each quote, it is important to provide some information about the person who reported it. You could include a pseudonym or other characteristic as it would add weight to your results and show the reader who said what.

• Need a section on limitations.

• Implications for practice and research – Based on the findings from this research, what are the implications for practice and research?

6. PLOS authors have the option to publish the peer review history of their article (what does this mean?). If published, this will include your full peer review and any attached files.

Reviewer #1: No

Reviewer #2: No

---

## [Author Response · Author response to Decision Letter 0]

29 Jun 2022

Dear the Editor and Reviewers,

We, the authors, would like to thank both the Editor and Reviewers for the time taken to thoroughly review and provide feedback to our manuscript. We have found this to be incredibly useful and enabled the manuscript to be developed into a more precise and well-thought through version.

Thank you for the opportunity to make these revisions to our manuscript and resubmit a version that addresses each of the points raised during the review process. The table in the attached "Response to Reviewers" letter shows each point raised by the Editor and each Reviewer, and how we have responded to each point. The line numbers stated are for the marked-up copy of the manuscript, so will differ to the unmarked version.

Thank you again for the invaluable feedback and time given, and for considering our manuscript for the PLOS ONE journal.

Yours sincerely, 

Kaylie Toll

On behalf of the authorship team

---

## [Decision Letter · Decision Letter 1]

14 Jul 2022

PONE-D-22-09068R1Consumer preferences, experiences, and attitudes towards telehealth: qualitative evidence from Australia.PLOS ONE

Dear Dr. Toll,

Thank you for submitting your manuscript to PLOS ONE. After careful consideration, we feel that it has merit but does not fully meet PLOS ONE’s publication criteria as it currently stands. Therefore, we invite you to submit a revised version of the manuscript that addresses the points raised during the review process.

We look forward to receiving your revised manuscript.

Kind regards,

Mukhtiar Baig, Ph.D.

Academic Editor

PLOS ONE

Journal Requirements:

Reviewers' comments:

Reviewer's Responses to Questions

**Comments to the Author**

1. If the authors have adequately addressed your comments raised in a previous round of review and you feel that this manuscript is now acceptable for publication, you may indicate that here to bypass the “Comments to the Author” section, enter your conflict of interest statement in the “Confidential to Editor” section, and submit your "Accept" recommendation.

Reviewer #1: All comments have been addressed

Reviewer #3: (No Response)

2. Is the manuscript technically sound, and do the data support the conclusions?

Reviewer #1: Yes

Reviewer #3: Yes

3. Has the statistical analysis been performed appropriately and rigorously? 

Reviewer #1: N/A

Reviewer #3: N/A

4. Have the authors made all data underlying the findings in their manuscript fully available?

Reviewer #1: Yes

Reviewer #3: No

5. Is the manuscript presented in an intelligible fashion and written in standard English?

Reviewer #1: Yes

Reviewer #3: Yes

6. Review Comments to the Author

Reviewer #1: (No Response)

Reviewer #3: The authors have not made all data available (but have explained the reasons for this adequately).

I felt the manuscript was well written and in the main, the previous reviewer comments had been dealt with adequately. Having not seen the manuscript prior to its resubmission following responses to peer review, I have not conducted a full, fresh review, but restrict my comments to the extent to which I feel that previous reviewer comments have been dealt with. I suggest some further minor amendments.

1. Reviewer 2 noted the possibility of coercion if participants were previously known to hosts or were social acquaintances. The authors did not address this point in their response but it is an important one that has bearing on the rigour, reproducibility and credibility of the data, so I would ask that the authors include a specific comment within their methodology on this. It is likely that most of the participants knew the hosts because the hosts took the lead on sampling and recruiting participants from within their networks. Allied to this - what influence (if any) did the researchers have on the diversity of participants that the hosts recruited e.g. was there a sampling frame that hosts were required to use, or did they have complete carte blanche over who they recruited to be in the KTD?

2. It is not clear why the issues with data recording arose in the first place. Usually, if a participant does not consent to being recorded, they are in essence not consenting to the study and ought to take no further part in it. It is strange that the authors chose instead to not record the entire session simply because one individual may not have wanted to be recorded. This may have implications for the quality of data from the sessions that were not recorded, and I would like to see more reflection on this approach (and its potential consequences) in the paper.

3. Whilst the authors have addressed a point about the data analysis process from the perspective of dual coding from a second researcher, the authors did not address the important point from reviewer 2 about member checking e.g. were hosts and/or participants given the transcripts so that they had the opportunity to amend anything they said or withdraw any of their statements? If not, why not?

Similarly, rigour and trustworthiness can be broader than simply having dual review of transcripts/coding (reviewer 2 point 12). I would invite the authors to consider how they ensured rigour/trustworthiness in a broader sense than just in the data analysis stage.

4. The authors chose not to add any information about the person saying a quotation because of the possibility of anonymity being broken. This is a legitimate concern but would actually not preclude the authors adding some very brief information on an individual that would not break participant confidentiality, such as gender, broad age (e.g. under 50, above 50), some sense of urbanity/rurality (given that this is one of the characteristics the authors surmise may make a difference to patient receptiveness to telehealth), or past use of telehealth yes/no. I would maintain that having some information about the person who reported a quotation is still very important, and can be done sensitively without any risk that a participant would be identified from the manuscript. I would ask the authors to reconsider their decision on this point.

7. PLOS authors have the option to publish the peer review history of their article (what does this mean?). If published, this will include your full peer review and any attached files.

Reviewer #1: No

Reviewer #3: No

---

## [Author Response · Author response to Decision Letter 1]

16 Aug 2022

Dear the Editor and Reviewers,

We, the authors, would like to thank both the Editor and Reviewers for the time taken to review our previous responses and provide feedback to our manuscript. We have found this to be incredibly useful and enabled the manuscript to be developed into a more precise and well-thought through version.

Thank you for the opportunity to make these revisions to our manuscript and resubmit a version that addresses each of the points raised during the review process. The table below shows each point raised by the Editor and each Reviewer, and how we have responded to each point. The line numbers stated are for the tracked changes copy of the manuscript, so will differ to the unmarked version.

Thank you again for the invaluable feedback and time given, and for considering our manuscript for the PLOS ONE journal.

Yours sincerely, 

Kaylie Toll

On behalf of the authorship team

---

## [Editor Report · Decision Letter 2]

18 Aug 2022

Consumer preferences, experiences, and attitudes towards telehealth: qualitative evidence from Australia.

PONE-D-22-09068R2

Dear Dr. Toll,

We’re pleased to inform you that your manuscript has been judged scientifically suitable for publication and will be formally accepted for publication once it meets all outstanding technical requirements.

Kind regards,

Mukhtiar Baig, Ph.D.

Academic Editor

PLOS ONE

Additional Editor Comments (optional):

No comments
---

## [Editor Report · Acceptance letter]

22 Aug 2022

PONE-D-22-09068R2 

Consumer preferences, experiences, and attitudes towards telehealth: qualitative evidence from Australia. 

Dear Dr. Toll:

I'm pleased to inform you that your manuscript has been deemed suitable for publication in PLOS ONE. Congratulations! Your manuscript is now with our production department. 

Kind regards, 

on behalf of

Professor Mukhtiar Baig 

Academic Editor

PLOS ONE